# The Second Life of Methylarginines as Cardiovascular Targets

**DOI:** 10.3390/ijms20184592

**Published:** 2019-09-17

**Authors:** Natalia Jarzebska, Arduino A. Mangoni, Jens Martens-Lobenhoffer, Stefanie M. Bode-Böger, Roman N. Rodionov

**Affiliations:** 1Division of Angiology, Department of Internal Medicine III, University Center for Vascular Medicine, Technische Universität Dresden, 01307 Dresden, Germany; natalia.jarzebska@uniklinikum-dresden.de; 2Department of Anaesthesiology and Critical Care Medicine, University Hospital Dresden, Technische Universität Dresden, 01307 Dresden, Germany; 3Department of Clinical Pharmacology, College of Medicine and Public Health, Flinders University and Flinders Medical Centre, Bedford Park 5042, Adelaide, Australia; arduino.mangoni@flinders.edu.au; 4Institute of Clinical Pharmacology, Otto-von-Guericke University, 39120 Magdeburg, Germany; jens.martens-lobenhoffer@med.ovgu.de (J.M.-L.); stefanie.bode-boeger@med.ovgu.de (S.M.B.-B.)

**Keywords:** asymmetric dimethylarginine (ADMA), symmetric dimethylarginine (SDMA), dimethylarginine dimethylaminohydrolase (DDAH), alanine:glyoxylate aminotransferase 2 (AGXT2), asymmetric α-keto-dimethylguanidinovaleric acid (ADGV), symmetric α-keto-dimethylguanidinovaleric acid (SDGV), Homoarginine, beta-aminoisobutyric acid (BAIBA), 6-guanidino-2-oxocaproic acid (GOCA)

## Abstract

Endogenous methylarginines were proposed as cardiovascular risk factors more than two decades ago, however, so far, this knowledge has not led to the development of novel therapeutic approaches. The initial studies were primarily focused on the endogenous inhibitors of nitric oxide synthases asymmetric dimethylarginine (ADMA) and monomethylarginine (MMA) and the main enzyme regulating their clearance dimethylarginine dimethylaminohydrolase 1 (DDAH1). To date, all the screens for DDAH1 activators performed with the purified recombinant DDAH1 enzyme have not yielded any promising hits, which is probably the main reason why interest towards this research field has started to fade. The relative contribution of the second DDAH isoenzyme DDAH2 towards ADMA and MMA clearance is still a matter of controversy. ADMA, MMA and symmetric dimethylarginine (SDMA) are also metabolized by alanine: glyoxylate aminotransferase 2 (AGXT2), however, in addition to methylarginines, this enzyme also has several cardiovascular protective substrates, so the net effect of possible therapeutic targeting of AGXT2 is currently unclear. Recent studies on regulation and functions of the enzymes metabolizing methylarginines have given a second life to this research direction. Our review discusses the latest discoveries and controversies in the field and proposes novel directions for targeting methylarginines in clinical settings.

## 1. Introduction

Despite significant advances in prevention and therapy, cardiovascular disease (CVD) remains the leading cause of morbidity and mortality worldwide, thus warranting the search for novel markers for early identification of individuals at high risk for cardiovascular complications and new therapeutic approaches to further decrease cardiovascular morbidity and mortality. Endogenous methylated analogues of the essential amino acid l-arginine were proposed more than 20 years ago as promising markers of increased cardiovascular risk and potential mediators of cardiovascular complications. The biological and clinical significance of endogenous methylarginines as cardiovascular markers was demonstrated in multiple epidemiological studies. At the same time, experimental studies demonstrated that at least in some pathological conditions, endogenous methylarginines can directly contribute to development of cardiovascular, metabolic and renal damage. Despite these promising findings the interest towards methylarginines eventually plateaued, presumably because of the lack of therapies that selectively lower their concentrations. However, the recent discovery of new functions of the enzymes that metabolize endogenous methylarginines has led to a renewed impetus in this field. The aim of this review is to discuss these discoveries and their therapeutic potential, as well as to address the key controversies and unanswered questions.

## 2. Endogenous Methylarginines as Cardiovascular Risk Factors

There are three different types of endogenous methylated analogues of l-arginine: (1) N^G^-monomethyl-l-arginine (NMMA), (2) symmetric N^G^,N’^G^-dimethyl-l-arginine (SDMA), and (3) asymmetric N^G^,N^G^-dimethyl-l-arginine (ADMA) (Figure 1) [1]. Two of these molecules, NMMA and ADMA, are able to competitively inhibit nitric oxide (NO) synthases with similar half maximal inhibitory concentrations (IC_50_) of about 2–3 μM [2,3]. Nitric oxide (NO) plays a key protective role in the cardiovascular system via multiple mechanisms including regulation of vascular tone, protection from oxidative stress, inhibition of platelets activation, modulation of smooth muscle cells proliferation and stimulation of angiogenesis. The discovery that both ADMA and NMMA inhibit NO production led to the hypothesis that their increased concentrations may facilitate cardiovascular damage. ADMA is present in blood at 5–10 times higher concentrations than NMMA [4,5], so most of the epidemiological studies focused on ADMA while the levels of NMMA were often not reported.

The association between elevated ADMA levels and endothelial dysfunction was first demonstrated by Vallance and colleagues in 1992 [4]. Subsequent studies showed that elevated plasma ADMA concentrations correlate with increased cardiovascular morbidity and mortality [6]. The list of the diseases and pathological conditions associated with increased ADMA concentrations includes, but is not limited to, hypercholesterolemia [7], hypertension [8], type 2 diabetes and insulin resistance [9], hypertriglyceridemia [10], coronary artery disease [11], stroke [12], peripheral arterial disease [13], chronic kidney disease [14] and preeclampsia [15].

In addition to being elevated in cardiovascular, renal and metabolic diseases, ADMA has been shown to be a strong predictor of cardiovascular complications and increased mortality in patients at risk. The Ludwigshafen Risk and Cardiovascular Health Study, a large prospective study with a median follow-up of 5.5 years, documented that plasma ADMA was independently associated with all-cause and cardiovascular mortality in patients with stable and unstable ischemic heart disease [16]. Similarly, Leong et al. found that elevated ADMA plasma concentrations were associated with increased incidence of myocardial infarction and stroke in women after 24 years of follow-up [17]. ADMA was also shown to be an independent predictor of mortality in patients with end-stage renal disease [18].

In their original work, Vallance and coauthors showed that the acute infusion of ADMA in healthy volunteers leads to endothelial dysfunction in the brachial artery [4]. Kielstein and colleagues demonstrated that infusion of ADMA in humans decreases renal blood flow and increases renovascular resistance, sodium retention and systemic blood pressure [19]. The same group demonstrated that ADMA infusion in healthy volunteers decreases total cerebral perfusion and increases arterial stiffness [20]. These studies suggest that ADMA may not only be a marker of cardiovascular disease, but can also play a direct role in cardiovascular damage in at least some of the pathological conditions associated with increased systemic ADMA concentrations.

The presence of endogenous NO synthase inhibitors such as ADMA may explain the “arginine paradox”, which refers to the apparently paradoxical observation that NO production increases in response to administration of l-arginine [21,22,23]. For example, infusion of l-arginine has been found to affect NO-dependent vasomotor responses in vivo, even though the normal concentration of l-arginine in plasma is 30-fold higher than the Michaelis–Menton constant (Km) for l-arginine of purified NO synthase [21,24]. This response to l-arginine was initially considered paradoxical because NO synthase was thought to be saturated with its substrate (l-arginine) so that further addition of l-arginine should not have affected the rate of NO production [23]. In the presence of ADMA, however, higher concentrations of l-arginine are needed to saturate NO synthase, providing a potential explanation for the observed “paradox” [23].

In contrast to ADMA and NMMA, SDMA (a structural analogue of ADMA [25]) does not directly inhibit nitric oxide synthase (NOS) [4]. Consequently, SDMA was initially thought to be biologically inert and was therefore included into the early epidemiological studies as a negative control. SDMA was shown to be a sensitive parameter of renal function, sometimes even more sensitive than creatinine [26]. Interestingly, a recent study showed that SDMA predicts all-cause mortality following ischemic stroke even after adjustment for renal function and novel risk factors such as C-reactive protein, albumin, beta-thromboglobulin, and the von Willebrand factor [27]. Furthermore, SDMA predicted all-cause-mortality in the general population after adjustment for traditional and novel cardiovascular risk factors, and renal function [28]. It is being currently investigated whether SDMA is indeed biologically inert or whether it exerts some physiological or pathophysiological effects. Consistent with the second possibility, it has been shown that SDMA may influence NO metabolism by interfering with l-arginine transport through cellular membrane by the cationic amino acid transporter 2B (CAT-2B) [29]. It has also been proposed that SDMA accumulates in high-density lipoprotein (HDL) fraction from patients with chronic kidney disease causing activation of toll like receptor 2 (TLR-2) and enhancing endothelial proinflammatory response [30].

## 3. Metabolism of Endogenous Methylarginines

To our current knowledge, there is no enzyme in mammals capable of directly methylating free l-arginine. Several enzymes, however, can methylate arginine residues that are incorporated into proteins. These enzymes are called protein arginine N-methyl transferases (PRMTs). The PRMTs transfer methyl groups from S-adenosylmethionine to arginine residues within proteins, generating S-adenosylhomocysteine as a byproduct. Subsequent proteolysis of the methylated proteins leads to the release of free ADMA, SDMA and NMMA into the cytoplasm (Figure 1) [31]. The substrates of PRMTs include but are not limited to RNA processing proteins, histones and transcription factors [31]. There are two types of PRMTs: type I and type II. Type I PRMTs monomethylate and asymmetrically dimethylate arginine residues within proteins, which leads to the formation of NMMA and ADMA, while type II PRMTs monomethylate and symmetrically dimethylate arginine residues within proteins, thus generating NMMA and SDMA [32,33].

The major routes of metabolism of endogenous methylarginines were described by Ogawa and colleagues [34]. After injection of radiolabeled ADMA into rats, the authors recovered 14% of the radioactivity in the urine and 3% in expired CO_2_, while the remainder accumulated in the tissues primarily in the form of citrulline. Most of the tissue radioactivity was found in the kidney, pancreas and liver, suggesting that these organs play a major role in ADMA metabolism. In contrast, after injection of radiolabeled SDMA into rats, the authors did not detect accumulation of radioactive citrulline in the tissues. In subsequent studies, Ogawa and colleagues identified the enzyme metabolizing ADMA to citrulline as dimethylarginine dimethylaminohydrolase (DDAH) [35]. Consistent with the results of the ADMA and SDMA infusion experiments, DDAH was shown to use ADMA, but not SDMA as a substrate. Currently the ADMA/SDMA ration is used as a marker of ADMA catabolism, with a high ADMA/SDMA ratio being suggestive of reduced ADMA excretion [36]. Another substrate of DDAH is NMMA [35].

Infusion of both radiolabeled ADMA and SDMA led to formation of the corresponding α-keto-derivatives: asymmetric α-keto-dimethylguanidinovaleric acid (ADGV, also abbreviated as DMGV) from ADMA and symmetric α-keto-dimethylguanidinovaleric acid (SDGV, also abbreviated as DM’GV) from SDMA. In subsequent studies, Ogawa and coauthors identified the enzyme responsible for transamination of endogenous methylarginines with formation of the corresponding α-keto-derivatives as alanine:glyoxylate aminotransferase 2 (AGXT2) [37].

The activities of DDAH and AGXT2 could explain the presence of all the observed radioactive metabolites of ADMA and SDMA except for their N-α-acetylated derivatives: asymmetric N-α-acetyl-N^G^, N^G^-dimethyl-l-arginine (N-α-acetyl-ADMA) and symmetric N-α-acetyl-N^G^,N’^G^-dimethyl-l-arginine [34,38]. N-α-acetylation has also been observed in humans as a minor pathway for the elimination of ADMA and SDMA [39,40]. Further studies are needed to identify the enzyme system responsible for N-α-acetylation of endogenous methylarginines and to assess its relative contribution to their metabolism (Figure 1).

## 4. Transport of Endogenous Methylarginines

The transport of NMMA, ADMA and SDMA across the plasma membrane is not well characterized. Studies in Xenopus laevis oocytes showed that all the three endogenous methylarginines can compete with l-arginine for the y+ cationic amino acid transporter 2B (CAT-2B) suggesting that this transporter may be involved in the uptake from an extracellular to intracellular environment of not only l-arginine, but also of NMMA, ADMA and SDMA [29]. It was also shown that at least ADMA can under certain conditions use several other cationic transport systems, such as cationic amino acid transporter 2 A (CAT2A), organic cation transporter 2 (OCT2) and multidrug and toxin extrusion protein 1 (MATE1) [41]. Further studies are needed to determine the relative contribution of each of these transport systems to the transport of endogenous methylarginines across the plasma membrane in vivo. In addition to transport across the plasma membrane, endogenous methylarginines could also be transported into mitochondria by the mitochondrial transporter SLC25A2 [42].

## 5. Dimethylarginine Dimethylaminohydrolases

The primary pathway for catabolism of ADMA and NMMA is via DDAH-catalyzed hydrolysis to citrulline (Figure 1) [35]. There are two isoforms of DDAH (DDAH1 and DDAH2) in mammals, each encoded by a separate gene [43]. Original analysis of tissue expression of human DDAH isoforms by Tran and colleagues suggested that DDAH1 was primarily expressed in the central nervous, gastrointestinal, respiratory, excretory and male reproductive systems, while DDAH2 was primarily expressed in blood cells, bone marrow, cardiovascular, gastrointestinal, respiratory, excretory and female reproductive systems [44]. Subsequent studies regarding the relative expression of different isoforms of DDAH in the cardiovascular system have been contradictory. Wang et al. used immunostaining to show that DDAH2 was the predominant DDAH isoform in mesenteric resistance vessels, while hardly any DDAH1 expression could be detected [45]. Dowsett and colleagues on the other hand demonstrated expression of DDAH1 in primary mouse pulmonary endothelial cells [46]. Zhang and coauthors reported expression of both DDAH1 and DDAH2 in human umbilical vein endothelial cells [47]. Schwedhelm and colleagues reported expression of both DDAH1 and DDAH2 in mouse hearts and aortas [48].

## 6. DDAH1 and ADMA

In order to determine the physiological and pathophysiological roles of DDAH isoforms, several groups generated genetically modified models with either upregulation or downregulation of DDAH expression. Dayoub and colleagues showed that transgenic overexpression of DDAH1 in mice decreases ADMA concentrations in plasma and tissue and increases NO production [49]. DDAH1 transgenic mice have increased basal NO production in the aorta and are protected from ADMA-mediated endothelial dysfunction in cerebral arterioles [50]. DDAH1 knockout mice have elevated plasma and tissue ADMA concentrations, endothelial dysfunction and hypertension [51,52]. Development of the experimental approaches for the manipulation of DDAH activity in vivo allowed testing of the potential role of ADMA/DDAH pathway in different mouse models of human diseases. Thus, it was shown that DDAH1 overexpression and ADMA lowering improves angiogenesis in the ischemic hind limb model [53] and protects from myocardial and renal ischemia-reperfusion injury [54,55]. Jacobi and colleagues demonstrated that DDAH1 overexpression ameliorated atherosclerosis in apolipoprotein E (ApoE)-deficient mice [56]. DDAH1 transgenic mice were protected from renal injury in a murine model of hypertension [57]. DDAH1 knockout mice were shown to have increased survival in septic shock, presumably due to ADMA-mediated inhibition of excessive NO production by inducible NOS (iNOS) [58].

Most of the existing evidence suggests that therapeutic strategies aimed at upregulation of DDAH activity and subsequent ADMA lowering might be beneficial to patients with CVD. As a result, several groups performed experimental screens for modulators of DDAH activity to identify DDAH-activators [59,60,61]. These screens led to discovery of specific DDAH inhibitors [62,63], however, no specific and efficient DDAH activators were reported. One of the potential explanations for this is that both DDAH isoenzymes are small proteins and, therefore, it is unlikely that they have any sites for allosteric activation. All the screens for DDAH activators, performed with the purified recombinant enzymes, have not been suitable for the identification of small molecules capable of regulating the DDAH isoenzymes at the level of post-translational modifications and protein–protein interactions.

## 7. Controversy 1: DDAH2 and ADMA

There is still a debate in the field about the relative contribution of DDAH2 towards the total DDAH activity. In contrast to the initial observation that DDAH2 is active towards ADMA in vitro, Pope and colleagues could not detect any significant increase in total DDAH activity in the lysate of endothelial cells with massive overexpression of DDAH2 even though those cells were producing higher amounts of NO [64]. Knockdown of *DDAH1* but not of *DDAH2* gene in cultured human umbilical vein endothelial cells led to increased accumulation of ADMA in the cell culture medium [52]. siRNA-mediated knockdown of *Ddah1*, but not of *Ddah2* gene led to increase in plasma ADMA concentrations in rats [45]. On the other hand Hasegawa and colleagues reported increased tissue DDAH activity in skeletal muscles and decreased plasma ADMA concentrations in mice with ubiquitous transgenic overexpression of mouse *Ddah2* gene [65]. DDAH2 transgenic mice were protected from ADMA- and angiotensin-II-induced oxidative stress, medial thickening and perivascular fibrosis in coronary microvessels [65]. Knockout of *Ddah2* gene in mice did not lead to changes in plasma ADMA concentrations, but nevertheless resulted in decreased ADMA concentration in urine, myocardium and kidneys as well as in impairment of endothelial function in aortic rings [66]. However, a potential confounding factor in this model might have been that *Ddah2* knockout mice also had decreased expression of *Ddah1* gene in the kidneys as compared to their wild type littermates, which could have affected ADMA metabolism independently of DDAH2 [66]. With the contradictory data on the role of DDAH2 in ADMA clearance, it is still unclear, which of the biological effects caused by modulation of DDAH2 expression in animal models are mediated by ADMA and which are ADMA-independent.

## 8. Non-Enzymatic Function of DDAHs

Both DDAH isoforms have been suggested to possess some nonenzymatic effects (summarized in Figure 2) in addition to their accepted main function of hydrolysis of ADMA and NMMA. Tokuo and colleagues demonstrated that DDAH1 can bind to tumor suppressor neurofibromin 1 (NF1), which is a negative regulator of the Ras oncogene pathway, and increases its phosphorylation at the cysteine/serine-rich domain by protein kinase A [67]. NF1 can regulate the proliferation and migration of vascular smooth muscle cells [68], so it is possible that altered NF1 function may have contributed to some of the vascular phenotypic effects seen in DDAH1 transgenic mice. The presence of ADMA-independent effects of DDAH1 is also supported by the observation that overexpression of the active site mutant of DDAH1 was able to affect the growth of glioma xenografts, while the total DDAH activity in the lysates of tumor cells remained unchanged [69]. Furthermore, downregulation of DDAH1 in a murine model of bleomycin-induced pulmonary fibrosis reduced collagen production by fibroblasts in the fibrotic lungs in an ADMA-independent manner [70]. Taking into account that overexpression of DDAH1 protected from hyperhomocysteinemia-induced hypertrophic changes in the vessel wall, but not from impairment in the NO-mediated endothelial function in our study, it is possible that the observed effects of DDAH1 in the vascular wall were at least partially ADMA- and NO-independent [71]. Hasegawa and colleagues demonstrated that, similar to DDAH1, DDAH2 can also participate in protein-protein interactions and can upregulate vascular endothelial growth factor (VEGF) expression in an NO-independent manner by increasing binding activity of SP1 transcription factor to VEGF promoter in a PKA-dependent manner [72]. Protein–protein interactions of DDAH2 might regulate its activity, which would provide an explanation for the described above observations that even massive overexpression of DDAH2 results in only mild, if any, increase in the total DDAH activity. Reversing potential negative protein-protein interactions of DDAH1 and DDAH2 may yield a novel therapeutic approach for upregulation of DDAH activity and treatment of ADMA-mediated pathologies.

## 9. Alanine: Glyoxylate Aminotransferase 2

Endogenous methylarginines can also be metabolized through an alternative pathway by AGXT2 with formation of the corresponding ketoacid derivatives: asymmetric α-keto-dimethylguanidinovaleric acid (ADGV, also abbreviated as DMGV) from ADMA, symmetric α-keto-dimethylguanidinovaleric acid (SDGV, also abbreviated as DM’GV) from SDMA (Figure 1) and α-keto-methylguanidinovaleric acid from NMMA [34,37]. It has been suggested that at least two of the α-keto-derivatives of endogenous methylarginines, namely ADGV and SDGV, can undergo further oxidation with formation of asymmetric dimethylguanidinobutyric acid (ADGB) and symmetric dimethylguanidinobutyric acid (SDGB), respectively [34]. When radiolabeled ADMA was injected into rats, most of the radioactivity recovered in the urine was in the form of either unmodified ADMA or the products of AGXT2-mediated hydrolysis of ADMA (ADGV and ADGB) [34]. This observation suggests that there are two main mechanisms for the renal clearance of ADMA: “direct”, when ADMA is excreted unchanged and “AGXT2-mediated”, when ADMA is first metabolized by AGXT2 and then excreted. A composite compound, consisting of both ADGV and SDGV has been recently recognized as an independent biomarker of CT-defined non-alcoholic fatty liver disease in the offspring cohort of the Framingham Heart Study participants and as a predictor of future diabetes up to 12 years before disease onset [73].

AGXT2 is a pyridoxal phosphate-dependent mitochondrial enzyme that is expressed in the liver and in the epithelial cells of Henle’s loop of the kidney [74,75]. It was discovered as early as in 1978 that mammals have two distinct isoenzymes possessing alanine-glyoxylate aminotransferase activity: alanine-glyoxylate aminotransferase 1 (AGXT1) and AGXT2 [76]. Both isoenzymes are able to catalyze the alanine-glyoxylate amino transfer reaction [76]. However, it has been shown that only the deficiency of AGXT1, but not of AGXT2, leads to primary hyperoxaluria type I in humans, suggesting that AGXT1, but not AGXT2, plays the major role in glyoxalate detoxification [77].

AGXT2 can also utilize D-beta-aminoisobutyric acid (BAIBA), endogenous nonproteinogenic amino acid, as an alternative amino donor [78]. There are two biologically active enantiomers of BAIBA: L-BAIBA and D-BAIBA, with presumably distinct physiological functions. L-BAIBA is produced from L-lysine [79], while D-BAIBA is an intermediate product of thymine catabolism [80]. The Km for D-BAIBA at the physiological pH is lower than the Kms of AGXT2 for the other amino donors, including L-alanine, suggesting that D-BAIBA may be the preferred AGXT2 substrate [81]. This would be consistent with the observation that patients with AGXT2 loss-of-function polymorphisms develop an autosomal recessive metabolic trait hyper-beta-aminoisobutyric aciduria, which is characterized by elevation of BAIBA concentration in urine and plasma [78]. The physiological role of BAIBA and the potential consequences of its elevations are still unclear, partially because in most of the experimental studies with BAIBA supplementation the animals were given a mixture of both D and B enantiomers. Recent studies, however, suggest that BAIBA may regulate lipid metabolism and induce browning of white adipose tissue through inducing gene expression of the mitochondrial uncoupling protein UCP-1, mitochondrial biogenesis transcription coactivator PGC-1α, and respiratory chain protein cytochrome c in a PPARα-dependent manner [82,83].

AGXT2 is also involved in the metabolism of some other bioactive substances. One of them is β-alanine, a nonproteinogenic amino acid which is a rate-limiting precursor of carnosine, a compound needed for high-intensity performance [84]. Interestingly, a recent genome-wide association study demonstrated that systemic concentrations of another nonproteinogenic amino acid, L-homoarginine, are elevated in patients with AGXT2 loss-of-function polymorphisms [85]. Recent epidemiological studies have demonstrated an association between low circulating concentrations of L-homoarginine and an increased risk of cardiovascular and all-cause mortality [86,87]. Our group showed that AGXT2 can metabolize homoarginine in vivo [88] and that homoarginine supplementation protects from cardiac remodeling in a mouse model of coronary artery disease [89]. The known substrates of AGXT2 are summarized in Figure 3.

It still remains to be established whether AGXT2 can play a direct role in the pathogenesis of diseases associated with decreased homoarginine concentrations.

Considerable clinical and experimental evidence suggest that reduction in NO availability might facilitate the development of chronic kidney disease (CKD) and kidney impairment [90,91,92]. In addition, experimental studies demonstrated that increased systemic ADMA concentrations are a powerful predictor of CKD independent of estimated glomerular filtration rate (eGFR) and other risk factors [18,93,94]. Decreased renal tubular ADMA metabolism was also shown to protect from worsening of renal function and ADMA was suggested to play both pathogenic and protective roles in different disease sites and time points [95,96]. Other groups showed that systemic SDMA concentrations are correlated with serum creatinine and eGFR, two key markers of renal function. Plasma SDMA is also elevated in patients with renal disease and increase progressively with worsening of renal function [93,94]. Recently, Bode-Böger et al. analyzed the role of homoarginine and 6-guanidino-2-oxocaproic acid (GOCA) (substrate and product of AGXT2, respectively) in patients suffering from non-dialysis chronic kidney disease. They showed that low plasma homoarginine concentrations and high GOCA concentrations are associated with worse prognosis. Elevated activity of AGXT2 in those patients could provide a possible explanation for this phenomenon, which may be a physiological reaction to elevated systemic ADMA and SDMA concentrations in those patients [97]. In addition, the AGXT2 rs37369 AA genotype was found to be significantly associated with elevated levels of blood urea nitrogen in chronic heart failure patients free from renal disease [98].

## 10. Controversy 2: AGXT2 as a Therapeutic Target

Our discovery of the role of AGXT2 in homoarginine metabolism further increases the complexity of the potential cardiovascular effects of AGXT2. On one hand upregulation of AGXT2 would be expected to have a beneficial influence on the cardiovascular system via lowering systemic concentrations of ADMA and SDMA. On the other hand, upregulation of AGXT2 may also result in some negative cardiovascular effects via lowering systemic concentrations of homoarginine and BAIB, with the latter being suggested to play a role in browning of adipose tissue, which is also thought to be beneficial for the cardiovascular system, and in mediating beneficial effects of exercise on endothelial function [99]. Another substrate of AGXT2 is beta-alanine, which is the rate-limiting precursor for the presumably cardiovascular protective antioxidant dipeptide carnosine [100]. While these opposite cardiovascular effects of different AGXT2 substrates might explain, why AGXT2 polymorphisms are so common in the general population, they might also make therapeutic targeting of AGXT2 challenging (Figure 4). However, it is still possible that therapeutic modulation of AGXT2 activity would be of a clinical benefit in selected groups of patients with pathologies mediated primarily by certain AGXT2 substrates with the other ones playing only a minor role. Solving the crystal structure of human AGXT2 should significantly advance the field and facilitate a better understanding of the regulation of substrate specificity of AGXT2 with the possible goal of identification of the small molecules or genetic modifications, which will increase substrates specificity of AGXT2 towards endogenous methylarginines, without affecting or decreasing its specificity to potentially “cardiovascular and metabolically beneficial” substrates.

## 11. Future Perspectives

The work presented in this review has significantly advanced our understanding of the multiple cardiovascular and metabolic roles of endogenous methylarginines and the enzymes involved in their metabolism. Furthermore, it has allowed identification of the novel therapeutic targets and created the basis for future translation of basic research findings into the clinical practice. The most promising research directions in this field include the further search for small molecules upregulating DDAH activity and the identification of the binding partners and regulatory posttranslational modifications of DDAHs, which could also be potentially targeted therapeutically. A better understanding of the regulation of AGXT2 activity with a specific focus on the role of the enzyme in liver and kidney disease and on the potential therapeutic approaches modulating the relative substrate specificity of AGXT2 is also of a significant importance in parallel to further exploration of the biological effects of its individual substrates.

## Figures and Tables

**Figure 1 ijms-20-04592-f001:**
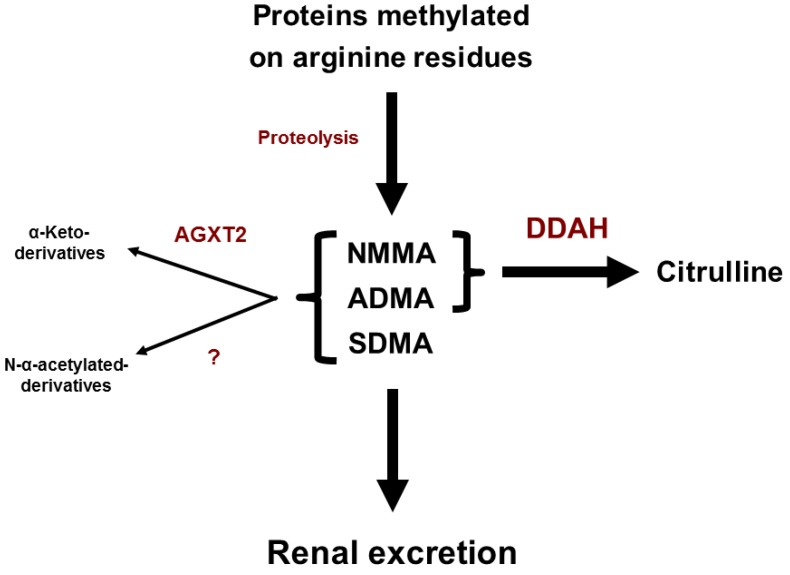
Metabolism of endogenous methylarginines. N^G^-monomethyl-l-arginine (NMMA), asymmetric N^G^,N^G^-dimethyl-l-arginine (ADMA) and symmetric N^G^,N’^G^-dimethyl-l-arginine (SDMA) derive upon hydrolysis of proteins methylated on arginine residues. ADMA and NMMA are further hydrolyzed to citrulline by dimethylarginine dimethylaminohydrolase (DDAH). All three endogenous methylarginines are also converted to the corresponding α-keto-derivatives by alanine:glyoxylate aminotransferase 2 (AGXT2). NMMA, ADMA and SDMA could be also N-α-acetylated, however, the enzyme, which is responsible for this reaction, is still unknown. All three endogenous methylarginines are excreted by kidneys.

**Figure 2 ijms-20-04592-f002:**
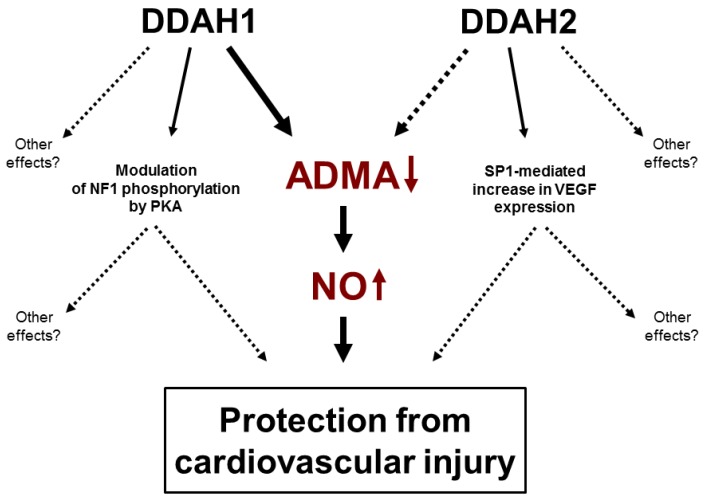
Protective effects of DDAHs. Solid lines—proven effects. Dashed lines—hypothesized effects. DDAH1 lowers ADMA, which leads to elevation in NO bioavailability and protects from cardiovascular injury. The relative contribution of DDAH2 to ADMA clearance is still not entirely clear. Both DDAH1 and DDAH2 have been shown to have ADMA-independent effects.

**Figure 3 ijms-20-04592-f003:**
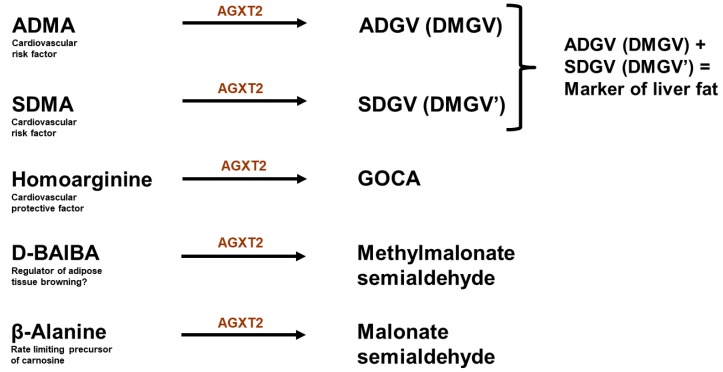
Biologically active metabolites of AGXT2. ADMA: asymmetric dimethyarginine; ADGV: asymmetric α-keto-dimethylguanidinovaleric acid; SDMA: symmetric dimethylarginine; SDGV: symmetric α-keto-dimethylguanidinovaleric acid; GOCA: 6-guanidino-2-oxocaproic acid; BAIBA: beta-aminoisobutyric acid. Alanine and glyoxylate (another substrates of AGXT2) are intentionally not included in the figure, as it is not clear whether AGXT2 plays a biologically significant role in regulation of their levels.

**Figure 4 ijms-20-04592-f004:**
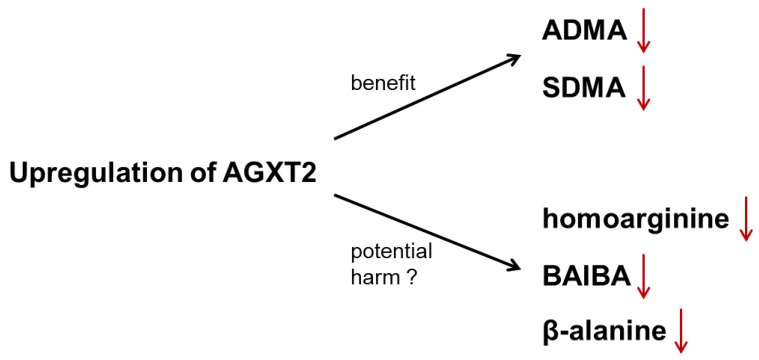
Janus-faced cardiovascular consequences of AGXT2 upregulation. Therapeutic upregulation of AGXT2 may exert a positive effect on cardiovascular system via lowering of ADMA and SDMA. However, there may also be a possible harmful effect caused by lowering of homoarginine, BAIBA and β-alanine levels.

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
