# Peer review of "The Second Life of Methylarginines as Cardiovascular Targets"

_ijms, 2019, doi:10.3390/ijms20184592_

Round 1

Reviewer 1 Report

The review written by Jarzebska and colleagues is a very well written and timely manuscript discussing the multiple cardiovascular and metabolic roles of endogenous methylarginines. The controversies regarding DDAH1/2 and AGXT2 are discussed well and the article has referred to appropriate articles as evidence of the statements made.  I particularly liked the figures and felt they were a good illustration of the effects/role of the enzymes involved in methylarginine metabolism. Overall a clear and concise review.

Author Response

We are very thankful for the time the reviewer took to read our manuscript and we are happy to receive positive recommendation.

Reviewer 2 Report

The Authors have written a narrative review regarding an interesting and still unrecognized topic, that is the possible involvement of methylarginines in the increased risk of cardiovascular disease appearance.

I have a few suggestions to improve their manuscript,namely:

Introduction: it is made by four main concepts (lines 34-37;lines 38-40; lines 40-44; lines 46-47). Each of them deserve a specific reference, please; Introduction: specify that SDMA is a structural isomer of ADMA. In this respect cite Bode-Böger SM, Scalera F, Kielstein JT, Martens-Lobenhoffer J, Breithardt G, Fobker M, et al. Symmetrical dimethylarginine: a new combined parameter for renal function and extent of coronary artery disease. J Am Soc Nephrol 2006;17(4):1128–1134; Page 3, line 132: added that the ratio ADMA/SDMA is a marker od ADMA catabolism. In this respect cite Bassareo PP, et al. Early Hum Dev 2014; 90: 173-176; Page 4, line: replace “in the transmembrane transport” with “in the ADMA uptake from an extracellular to intracellular environment”, please;

Author Response

We are very thankful for the comments of the reviewer. According to her/his suggestions we made the following changes in the text of the manuscript:

We added the information that SDMA is a structural analogue of ADMA (line 97) We added the information that the ADMA/SDMA ration is used as a marker of ADMA excretion (line 133-134) We changed 'in the transmembrane transport' to 'in the ADMA uptake from an extracellular to intracellular environment' (line 161-162)